# Deblurring for spiral real-time MRI using convolutional neural networks

**Yongwan Lim**                                                          YONGWANL@USC.EDU

**Shrikanth S. Narayanan**                                               SHRI@EE.USC.EDU

**Krishna S. Nayak**                                                     KNAYAK@USC.EDU

*Ming Hsieh Department of Electrical and Computer Engineering*

*University of Southern California, Los Angeles, CA, USA*

## Abstract

Spiral acquisitions are preferred in real-time MRI because of their time efficiency. A fundamental limitation of spirals is image blurring due to off-resonance, which degrades image quality significantly at air-tissue boundaries. Here, we demonstrate a simple CNN-based deblurring method for spiral real-time MRI of human speech production. We show the CNN-based deblurring is capable of restoring blurred vocal tract tissue boundaries, without a need for exam-specific field maps. Deblurring performance is superior to a current auto-calibrated method, and slightly inferior to ideal reconstruction with perfect knowledge of the field maps.

**Keywords:** Artifact correction, CNN, deblurring, off-resonance, real-time MRI

## 1. Introduction

Spiral acquisitions are widely used in real-time magnetic resonance imaging (RT-MRI) due to their time efficiency (Meyer et al., 1992), which has made it possible to capture vocal tract dynamics during natural speech (Lingala et al., 2016a). A fundamental limitation of spirals is image blurring due to off-resonance (Block and Frahm, 2005). This blurring is spatially varying and degrades image quality most significantly at air-tissue boundaries (Schenck, 1996), which are the exact locations of interest in speech RT-MRI.

Several methods exist to resolve this artifact (Nayak et al., 2001; Sutton et al., 2010; Lim et al., 2019a), but most of them 1) are computationally slow and 2) require explicit knowledge of *field maps* at the cost of scan time efficiency. Convention methods with these limitations hardly resolve the artifact for RT-MRI applications where low-latency processing and high temporal resolution are crucial. Recently, convolutional neural network (CNN) approaches have shown promise in solving this specific deblurring task (Zeng et al., 2019; Lim et al., 2019b). Compared to conventional methods, CNN would learn from training samples to recognize and undo the characteristic effects of off-resonance. Once trained, the feedforward operation of CNN is fast and does not rely on measuring exam-specific field maps but can generate a desired sharp image given a blurry image input in an end-to-end manner.

In this work, we utilize a simple CNN-based method to restore blurred vocal tract articulator boundaries for spiral RT-MRI of speech production. We consider this application because off-resonance appears as spatially (and temporally) abruptly varying blur at the articulators of interest and is a fundamental limitation to address. We evaluate the proposed method quantitatively using image quality metrics and qualitatively via visual inspection, and via comparison with conventional deblurring methods.

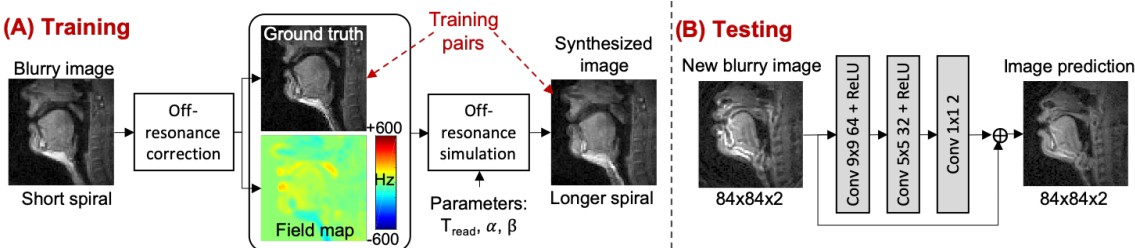

Figure 1: (A) Generation of training data. (B) 3-layer residual CNN architecture

## 2. Materials and Methods

**Training data** We used 2D spiral RT-MRI data acquired from 33 subjects (each consisting of 400 frames with an image size of $84{\times}84{\times}2$) on a 1.5 Tesla scanner using a vocal-tract imaging protocol (Lingala et al., 2016b). Ground truth images and field maps were obtained after applying a recently proposed off-resonance deblurring method (Lim et al., 2019a) (Figure 1A). Distorted images were simulated from the ground truth images with augmented field maps $\mathbf{f}'(x, y, t)$ frame-by-frame using two spiral trajectories with short and long readout durations ($T_{read}$) of 2.52 and 7.936 ms, respectively. We augmented the field maps by $\mathbf{f}' = \alpha\mathbf{f} + \beta$ to simulate varying range of off-resonance and global offset. Note that the severity of image blurring would be proportional to $|T_{read} \cdot \mathbf{f}'|$. We split data into 23, 5, and 5 subjects for training, validation, and test sets, respectively. We tested the impact of augmentation strategies on performance in a variety of settings (not shown here).

**Network architecture** The network is composed of 3 convolutional layers (Figure 1B). The first 2 layers apply 2D convolution, followed by ReLU operation. The last layer applies 2D convolution with $1{\times}1$ spatial support only. We add a skip connection. Both the input and output of the network consist of two channels (real and imaginary). The number of filter width was set to 64, 32, and 2. We choose the filter sizes of 9, 5, and 1, based on performance in terms of image quality metrics using the validation set. The model is trained using the training set with a combination of L1 loss and gradient difference loss (Mathieu et al., 2016).

## 3. Experiments and Results

**Evaluation on synthetic test data** We applied multi-frequency interpolation (MFI) (Man et al., 1997), model-based iterative reconstruction (IR) (Sutton et al., 2003), and the proposed CNN into synthetically generated test data. It should be noted that we assume a reference field map to be known for both the MFI and IR methods although it is not available in practice. All those methods deblurred images frame-by-frame. In Figure 2, blurring is observed around the lips, tongue surface, and soft palate in the uncorrected image. After deblurring, MFI even deteriorates the delineation of boundaries in those regions, whereas the IR method almost perfectly resolves the blurring artifact, which is also shown in Figure 2B and on quantitative quality measures. The proposed method successfully resolves the blurring artifact in those regions, which is visually comparable to the result from IR. Both IR and the proposed methods exhibit sharp boundaries between the tongue and air and around the soft palate, which is found consistent over time (yellow arrows, Figure 2B).

**Evaluation on experimental in vivo data** We applied the trained network to real data acquired using the short and long spiral acquisitions. Note that the long spiral acquisition

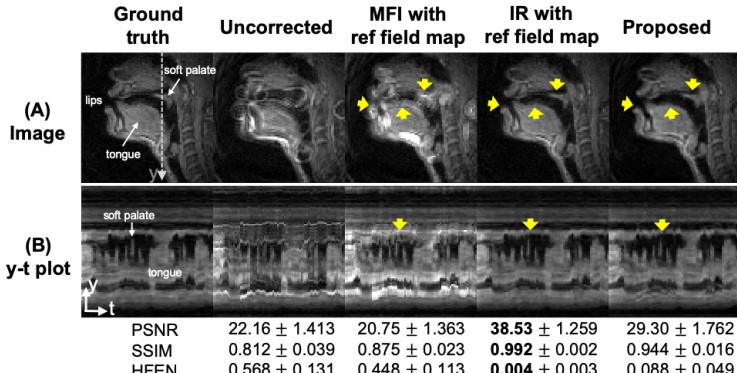

| | Ground truth | Uncorrected | MFI with ref field map | IR with ref field map | Proposed |
|---|---|---|---|---|---|
| PSNR | | 22.16 ± 1.413 | 20.75 ± 1.363 | **38.53** ± 1.259 | 29.30 ± 1.762 |
| SSIM | | 0.812 ± 0.039 | 0.875 ± 0.023 | **0.992** ± 0.002 | 0.944 ± 0.016 |
| HFEN | | 0.568 ± 0.131 | 0.448 ± 0.113 | **0.004** ± 0.003 | 0.088 ± 0.049 |

Figure 2: Representative synthetic images from the long spiral acquisition. (A) Images before and after deblurring with various methods. (B) Intensity vs time plot marked by the dotted line in (A). Mean and standard deviation of PSNR, SSIM, HFEN are present below, averaged over time and across readouts and subjects.

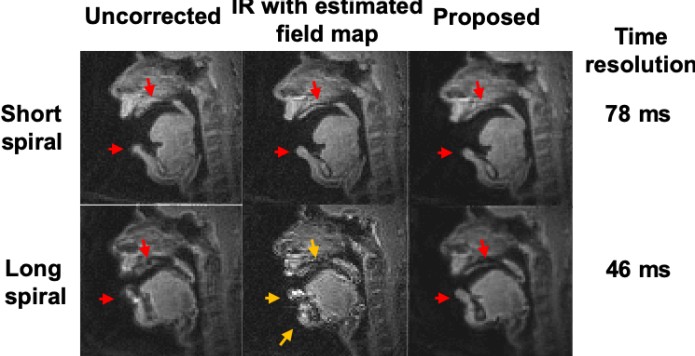

Figure 3: Representative experimental results using the short and long spiral acquisitions for comparison methods. Note that the short spiral uncorrected image represents one of the current standard practice (Lingala et al., 2016b).

enables 1.7-fold higher scan efficiency than the short one (a current standard practice), but inherently experiences much severe blurring (see red arrows in Figure 3). We compared results with an IR-based method (Lim et al., 2019a). This auto-calibrated method estimates dynamic field maps from distorted images itself with no scan time penalty. As shown in Figure 3, the proposed method provides improved delineation of the boundaries, which is consistent for both the short and long spiral acquisitions, whereas the IR method does not reliably work at the longer spiral acquisition (yellow arrows). This is a practical limitation of conventional methods where performance is limited by the quality of field map estimates.

## 4. Conclusion

We demonstrate an efficient, field-map-free approach for deblurring of spiral speech RT-MRI. The CNN-based deblurring is shown to be effective at resolving spatially varying blur at the vocal tract articulator boundaries, even with the longer spiral acquisition. In the context of speech RT-MRI, this can enable 1.7-fold improvement in scan efficiency. The proposed method is also computationally fast (12.3 ± 2.2 ms per-frame), enabling low-latency processing for RT-MRI applications.

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
