# OpenReview forum: "Deblurring for spiral real-time MRI using convolutional neural networks"
_MIDL.io/2020/Conference — MIDL 2020_

### Official Review · AnonReviewer2 · 2020-03-08
**A well structured paper dealing with deblurring for spiral real-time MRI by residual networks trained on synthetic data**

**Rating:** 4
**Confidence:** 3

**Review:**

(+) the paper is nicely written and easy to follow
(+) the idea of synthesizing data with blurring related to long spiral acquisition for subsequent training of a residual network for artifact reduction is reasonable
(+) the graphical abstract in Figure 1 is neat. (Note that the . is missing in the caption)
(+) evaluation is performed on both, synthetic and real test data

(-) evaluation on the synthetic test data is limited to MFI and IR based on the reference field map, only.
A comparison of your method to MFI and IR with estimated field maps would be interesting.
(-) the runtime evaluation (12.3 ms per frame)  provided in the conclusion should be part of the experiments section.
Please note duration of the comparative methods as well.

I recommend acceptance of this short paper.

---

### Official Review · AnonReviewer1 · 2020-03-11
**A CNN-based method is introduced to deblur MRI with spiral samplings.**

**Rating:** 3
**Confidence:** 5

**Review:**

Sprial sampling of MRI data is very time efficient may requires special reconstruction algorithms to reduce artifacts.
This method introduced a CNN-based method to deblur spiral-sampled MRI data.
A novel method was introduced to synthesize distorted data with augmented field maps.
But the IR with ref field map seems to have better performances than the proposed method.

---

### Official Review · AnonReviewer3 · 2020-03-14
**Brief presentation of original work on an interesting application**

**Rating:** 4
**Confidence:** 3

**Review:**

The authors show that a CNN deblurring approach gives promising results for real-time spiral readout MRI. The work is reasonably convincing, as a first step, and achieves impressive-seeming results. Clarity is good, with the authors explaining both their problem and proposed solution. Originality lies mainly in the application: there does not seem to be a lot of pre-existing work in the literature. Significance seems sufficient for a short format paper – there may be practical issues with the authors' approach compared to reference approaches, but this is a promising first step. On balance, I recommend this paper for acceptance.

Pros:
* Original work on an interesting application.
* Convincing evaluation of performance.
* Well-written and clear.

Cons:
* Unclear to what extent Figure 3 is truly representative – it might be beneficial to include several example images.
* Limited discussion of the quantitative results in Figure 2.
* No quantitative results for experimental data (due to lack of ground truth). Understandable in a short paper format, however.

---

### Official Review · AnonReviewer4 · 2020-03-14
**a promising method**

**Rating:** 4
**Confidence:** 4

**Review:**

When there is no ground truth, it is difficult to directly train a CNN to do deblurring.
This paper use delurring result from another method, and simulate blurred input using a physical model.
In this way, the paired image follows the physical model of blurring.

Pros:
The method respects the physics in MRI. And the result is very promising.

Cons:
(Minor) The first experiment did not compare with Lim et al., 2019a.

---

### Meta-Review · Area_Chair1 · 2020-03-27
**MetaReview of Paper47 by AreaChair1**

**Rating:** 3

**Metareview:**

This paper applies CNN to do deblurring for spiral real-time MR imaging. The current presentation is okay for a short paper.

**Paper Type:**

validation/application paper

---

### Decision · Program_Chairs · 2020-04-11

Accept